# Effects of Aquatic Interventions on Physical Health Indicators in People with Intellectual Disabilities: A Review

**DOI:** 10.3390/healthcare11141990

**Published:** 2023-07-10

**Authors:** George Tsalis, Glykeria Kyriakidou

**Affiliations:** School of Physical Education and Sport Science at Serres, Aristotle University of Thessaloniki, 62110 Serres, Greece; glikakiriakidou@hotmail.com

**Keywords:** intellectual disabilities, aquatic interventions, swimming, fitness, health

## Abstract

Exercise in water is widely used for improving the physical capacities of people with intellectual disabilities (IDs). In this review, we examined the results of studies on the effects of water interventions on functional abilities, as well as the physiological and biochemical status of people with IDs. We considered studies in the PubMed, MEDLINE, Scopus, Google Scholar, and Research Gate databases which were published between 2010 and 31 December 2022. Based on our inclusion criteria, 15 studies were selected for review. We found incomplete recording of data on the intensity and timing of physical activity. There was also wide variation in the terminology used to describe physical abilities. The study results suggested that aquatic interventions brought about improvements in endurance and strength; however, conflicting data were found with respect to balance and body mass index, and there were few data on biomarkers related to stress and brain health. Further research with more accurate training load data and a more common fitness terminology is needed. Lastly, traditional swimming should probably be considered as an aquatic intervention.

## 1. Introduction

Intellectual disabilities (IDs) are characterized by significantly below-average cognitive functioning, with an IQ < 70 [1]. The term ID is applied to a very heterogeneous group of individuals [2], and four clinical levels of severity exist: mild, moderate, severe, and profound, including Down syndrome (DS) and other unspecified IDs [3].

People with IDs experience physical problems exacerbated by low fitness levels, including functional parameters such as strength, endurance, coordination, and balance. The lack of physical abilities makes it difficult for these people in their daily lives, such as moving around and looking after themselves. In addition, many health problems occur at high rates in people with intellectual disability, such as specific syndromes, for example, congenital heart defects, poor mental health, and psychosocial disturbances. For these reasons, in addition to physical condition, other healthcare indicators, such as biological and biochemical parameters, should be considered when monitoring the vital status of people with IDs. Data on body composition, heart rate, blood lipids, and hormones are all necessary to fully evaluate biological function in such individuals [4,5].

It is well known that regular physical activity contributes to maintaining and improving the functional abilities in people with or without IDs [6,7]. Regular exercise of sufficient duration (≥12 weeks), intensity (≥50 VO_2_ peak), time (≥45 min), and frequency (≥2 times/week) significantly improves cardiovascular fitness and exercise capacity; however, such exercise seems to be of limited effectiveness with respect to anthropometric and body composition variables [8].

One frequently suggested intervention involves physical activity in an aquatic environment. There are many activities which can be performed in water, including Structured Water Dance Intervention (SWAN) [9], walking, standing movements [10], adapted aquatics, aquatic therapy, aquatic exercise, recreational aquatics, the use of small craft, and water-safety training [11], as well as conventional swimming [11,12]. Swimming is an effective form of exercise which provides several health benefits to humans [13,14,15]. However, while individuals in the general population typically swim proficiently, with good technique and appropriate levels of intensity, people with IDs often have difficulties with swimming [6]. In addition, in other forms of aquatic interventions, most exercise routines are similar to those applied on dry land, such as dance or calisthenics. This raises the possibility that a program can be standardized when parameters such as training volume and intensity can be controlled. In this review, we sought to document the types of aquatic activity—particularly swimming—reported by researchers in recent years, and how such activities impact the functional abilities of people with IDs, as well as their physiological and biochemical status.

## 2. Materials and Methods

### 2.1. Research Selection Strategy

The following index databases were used to obtain the appropriate literature for review: PubMed/MEDLINE, Scopus, Google Scholar, and Research Gate. Studies were collected until 31 January 2023. The research question was developed using the Population, Intervention, Comparison, Outcome (PICO) framework. We wanted to investigate, in the literature, whether people of any age with intellectual disabilities participate in aquatic activities, particularly in swimming, and whether these programs affect their functional abilities and physiological and biochemical status. Is there a positive effect, and if so, which programs are conducive to this effect? During the search, the following terms and combinations of terms were used: “intellectual disability”, “mental retardation”, “aquatic exercise”, “swimming”, “physical activity», «strength”, “power”, “balance”, “body composition”, “fat mass”, “hematological parameters”, “biochemical parameters”, “stress”, and “benefits”. Each combination of keywords was required to have the first four terms with the OR disjunction between them in pairs and the AND conjunction between the pairs. In total, there were six combinations (Table 1).

### 2.2. Inclusion Criteria

A study was included in our analysis if it satisfied the following criteria: (a) it was published in English; (b) the participants were people diagnosed with ΙD and/or DS; (c) the research involved an experimental group; (d) the research involved experimental studies; (e) the research concerned a specific aquatic education program; (f) it was published between 2010 and 31 December 2022.

### 2.3. Exclusion Criteria

A study was excluded from our analysis if it satisfied any of the following criteria: (a) it was not published in English; (b) the condition of participants did not meet qualifying criteria for the category of intellectual disability; (c) it was a case study; (d) there were no quantitative data for the educational process described; (e) the studies had an abstract only, apart from the conference abstracts.

### 2.4. Data Extraction and Selection

A review of the available literature was performed in accordance with the guidelines of Preferred Reports for Systematic Reviews and Meta-Analysis (PRISMA) (Figure 1). The information extracted from each study was as follows: author name or names; year of publication; the number of participants involved; their gender, age, and type of ID; the kind of training intervention; training details; outcomes.

## 3. Results

When considering aquatic interventions, it is sometimes difficult to clarify what is meant by the word “swimming”. Most aquatics programs use the term in their descriptions, but swimming itself is typically presented as a part of the intervention, and few details are provided about how to swim. Flotation equipment is often used to aid buoyancy or simply to promote a sense of fun. Table 2 summarize studies on aquatic interventions published over the past decade. Study results are presented in chronological order, from the most recent to the oldest. Table rows show author names, the year of publication, the number of participants and their disability, the age of participants, the kind of intervention studied and its frequency, the analytical program involved, and the functional, physiological, and biochemical outcomes reported.

In seven studies, [5,6,12,16,17,18,19], only regular swimming was involved. In five studies, swimming formed a part of an aquatic intervention program [20,21,22,23,24]. Lastly, in three studies, aquatic intervention alone was carried out [9,10,25].

Most of the studies in our review described swimming programs which focused only on functional capacity and biological parameters; only one study considered parameters of blood fat and immune function [17].

Interventions of 8–33 weeks in duration, involving 30–90 min of swimming, 2–3 times a week, led to improvements in aerobic capacity, endurance, muscular strength, balance, and functional ability, as well as improved times for swimming a 12 m distance [6,12,17]. However, there were no effects on limb speed, shoulder mobility, or times for swimming 24 m [6,12]. Contradictory results were found for flexibility. One study showed an improvement [17], but others found it to be unchanged [6,12]. Researchers also found that, after a 10–12-week period of swimming sessions lasting 45–60 min, 2–3 times per week, cardiorespiratory fitness, strength, jumping from standing, running speed, water skills, motor accuracy, motor integration, manual dexterity, balance, coordination, and agility were all unaffected [16,18].

The authors of [6,12] found that swimming for 30–90 min, 3 days per week, for 8–33 weeks, resulted in reduced body mass. However, other researchers [18,19] found that swimming for 60 min, 3 days per week, for 10–16 weeks did not affect body mass [18,19]. Similarly, some studies found that swimming for 50–90 min, 3 days per week, for 33–36 weeks resulted in decreased BMI [5,6,12], while other studies concluded that swimming for 45–60 min, 2–3 days per week, for 12–16 weeks, did not affect BMI values [16,17,19].

More contradictory results were found in relation to body fat. The authors of [5,6] found that, after swimming for 70–90 min, 3 days per week, for 33–36 weeks, body fat levels decreased. However, other researchers reported that, after 60 min of swimming, 3 days per week, for 16 weeks, body fat increased [19]. Lastly, the authors of [17,22] concluded that, after 60 min of swimming, 2 days per week, for 16 weeks, body fat remained unchanged [17,22].

Another study found that swimming for 70–90 min, 3 days per week, for 33 weeks, increased levels of VO_2_ max, while HR max was not affected [6]. When aquatic interventions, adapted swimming, and swimming were brought together as a combined exercise, the results showed that, after training for 35–60 min, 2–3 days per week, for 6–14 weeks, aerobic capacity and functional ability both increased [10,25], although speed remained unaffected [10,23,25]. Endurance, strength, balance, and mobility either improved or remained unaffected [10,23,25]. After aquatic training or adapted swimming for 35–60 min, 3 days per week, for 6 weeks to 6 months, vital capacity and cardiovascular parameters both increased [20], whereas body mass, BMI, body fat, and heart rate were all unaffected [18,22,23,25].

The biochemical profile of individuals with IDs does not differ significantly from that of the general population, except for some parameters that may be influenced by their diet [26,27]. Swimming for 60 min, 2 days per week, for 16 weeks, did not seem to significantly affect biochemical parameters such as TC, LDL-C, HDL-C, IgA, or IgM. However, one study reported increased levels of TG and IgG [17]. After aquatic training for 40–50 min, 1–3 days per week, for 12 weeks, levels of salivary cortisol, BDNF, and VEGF all decreased [9,21] while IGF-1 remained unaffected [21].

**Table 2 healthcare-11-01990-t002:** Descriptive characteristics of the studies that proposed aquatic intervention.

Authors	Participants	Activity	Features of Exercise Programs	Functional Ability	Biological Parameters	Biochemical Parameters
Suarez-Villadat B.; Oliva L.L.; Acebes C.; Villagra A. (2020) [5]	8 M + 7 F with DS 14.30 ± 1.25 years	Swimming, 50 min, 3 days/week, 36 weeks	First stage (10 min): swimming in front crawl style and breaststroke style. Second stage (35 min): exercises using technical support elements (e.g., pullbuoy, tables, shovels, and fins) increasing the heart rate to between 140–160 beats/min; swimming in front crawl and breaststroke, maximum heart rate 160–180 beats/min in front crawl; lowering the swimming intensity in breaststroke technique, heart rate 110–130 beats/min. Third stage (5 min): soft swim, muscle relaxation, and lowering heart rate and respiratory rate. The back- or breaststroke technique was used in this phase. Each session consisted of 800 and 1000 m.		BMI ↓ body fat ↓	
Boer P. H. (2020) [12]	13 M + F with DS 34.2 ± 5.0 years	Swimming, 30 to 40 min, 3 days/week, 8 weeks	Warmup (4 min): walking in circular motion inside the pool (1 min); marching in place whilst swinging the arms (1 min); a few simple stretches (single-arm crossover, chest stretch, and hamstring, calf, and quad stretch) (2 min). Preparation for main session (6–7 min): two intervals of high-intensity running on the spot (2 × (1 min interval, 30 s rest)); one set of lunge jumps (45 s, 15 s rest); one set of squat jumps (45 s, 15 s rest); flutter kicks whilst holding onto the side of the pool (1 min, 30 s rest). Main session (20 min): repetitive freestyle swim training (17 min); swimming lengths whilst holding onto kicking board (3 min).	Aerobic capacity ↑ endurance ↑ dynamic balance ↑ muscular strength ↑ 12 m swim time ↑ functional ability ↑ shoulder flexibility ↔ 24 m swim time ↔	Body mass ↓ BMI ↓	
Naczk A.; Gajewska E.; Naczk M. (2021) [6]	7 M + 4 F with DS 14.9 ± 2.35 years	Swimming, 70 to 90 min, 3 days/week, 33 weeks	Warmup outside the pool (10 min). Warmup in water (10 min). The main part of the training session lasted 30 min for the first 4 weeks, 40 min for the next 4 weeks, and 50 min for subsequent weeks. The recovery part (games in the water) lasted 20 min. During these training sessions, the adolescents usually swam 700–1000 m. At this stage, the swimming session lasted 90 min.	Aerobic capacity ↑ speed of limb ↔ handgrip ↑ balance ↑ flexibility ↓↔ sit-ups ↑ arms strength ↑ arms endurance ↑	Body mass ↓ BMI ↓ body fat ↓ VO_2_max ↑ HRmax ↔	
Lundqvist L-O.; Matérne M.; Frank A.; Mörelius E.; Duberg A. (2022) [9]	20 M + 14 F with ID 33.8 (21–53) years	Aquatic training, 40 min, 1 day/week, 12 weeks	Each session followed a structured program focusing on the following key components: experience of dance and music; adapted movements; stimulation of the senses; social interaction. A playlist of nine music tracks was created to stimulate the rhythm of the movements performed, to support relaxation, and to elicit different emotions.			Salivary cortisol ↓
Kim, H.; Lee, A.; Oh, J. (2018) [17]	10 children with ID 14.23 ± 9.87 years	Swimming, 60 min, 2 days/week, 16 weeks	Warmup (10 min): stretching and walking, 40–50% HRR (7–9 RPE). Main exercise (40 min): weeks 1–8, 50–60% HRR (9–11 RPE), kick, respiration, pull, combination, freestyle swim; weeks 9–16, 60–70% HRR (11–13 RPE), kick, respiration, pull, combination, drill, backstroke swim. Cooldown (10 min) 40–50% HRR (7–9 RPE).	Muscular strength ↑ cardio vascular endurance ↑ flexibility ↑ muscular endurance ↑	Body fat ↔ BMI ↔	TC ↔ TG ↑ LDL-C ↔ HDL-C ↔ IgG ↑ IgA ↔ IgM↔
Pérez C.A.; Carral J. M.C.; Costas A.Á.; Martínez S.V.; Martínez-Lemos R.I. (2018) [16]	7 M + 7 F with DS 37.07 ± 7.34 years	Swimming, 45 min, 2 days/week, 12 weeks	Warmup (15 min): inspiration/expiration inside the water. Sets: 3 R × 30 s × 2 sets. RBR: 5–10 s/RBS: 1 min, SS: medium. Timing: 5 min. Crawl kicks while holding the edge of the pool. Sets: 3 R × 30 s × 2 sets. RBR: 10 s/RBS: 1 min. SS: medium-high. Timing: 5 min. Crawl kicks while holding the edge of the pool. Sets: 3 R × 30 s × 2 sets. RBR: 10 s/RBS: 1 min. SS: medium-high. Timing: 5 min. Main part (30 min): crawl stroke (arms movement technique with pullbuoy). Sets: 2 R × 15 min × 3 sets. RBR: 10 s (PR in shallow water). RBS: 1 min (active rest: static trot and arms movement). SS: high. Timing: 10 min. Crawl stroke (legs movement technique). Sets: 2 R × 15 min × 3 sets. RBR: 10 s (PR in shallow water). RBS: 1 min (active rest: static trot and arms movement). SS: high. Timing: 10 min. Backstroke (legs movement technique). Sets: 2 R × 15 min × 3 sets. RBR: 10 s (PR in shallow water). RBS: 1 min (active rest: static trot and arms movement). SS: high. Timing: 10 min. Cooldown (5 min). Participants stayed in a higher temperature pool with ludic elements or in the whirlpool bath area.	Handgrip strength ↔ standing jump ↔ running speed ↔ cardiorespiratory fitness ↔ aquatic skills ↔	BMI ↔	
Boer P. H.; de Beer Z. (2019) [25]	8 M + 5 F with DS 31.2 ± 6.9 years	Aquatic training 35 to 45 min, 3 days/week, 6 weeks	Warmup (5 min), core 35 min session, and a 2 min cooldown. The exercises included arm circles, side twists, walking in place, running in place, water scoops, side leg lifts, flutter kicks on back, flutter kicks on stomach, jumping jacks, knee twists, side shuffles, squat jumps, lunge jumps, and a longer jog in place.	Aerobic capacity ↑ functional ability ↑ muscular strength ↑ balance ↔	Body mass ↔ BMI ↔	
Top E. (2015) [18]	6 M + 8 F with ID 17.4 ± 1.6 years	Swimming, 60 min, 3 days/week, 10 weeks	During 10-week swimming exercises, activities performed by the individuals with ID were prepared according to the Special Olympics Swimming Guide and other resources about swimming. Exercises were performed under supervision of 20 helpers and a head coach with swimming training.	Motor precision ↔ motor integration ↔ manual dexterity ↔ bilateral coordination ↔ balance ↔ jumping speed ↔ agility ↔ upper-limb coordination ↔ strength ↔	Body mass ↔	
Pehoiu C.; Moacă G.M.; Stănescu F.M. (2015) [20]	8 M + 4 F (5 with ID) 7–19 years	Adapted swimming, 50 min, 3 days/week, 6 months	Special swimming and aquagym materials (rafts, palm and feet swimming, mask snorkel tube type, pullbuoy, stick figures and floating circles, balls, etc.). (A) Basic swimming skills in the water, such as flotation, breathing, and propulsion. (B) A large number of exercises and repetitions, exercises of mild to moderate intensity. (C) Increased intensity, shorter breaks between exercises or games, and higher speed of execution.		Vital capacity ↑ cardiovascular parameters ↑	
Hakim R.M.; Ross M.D.; Runco W.; Kane M.T. (2017) [10]	13 M + 9 F with ID 37.14 ± 9.45 years	Aquatic training, 45 to 60 min, 2 days/week, 8 weeks	Warmup: lap walking × 4 laps each, walking forward and backward, sidestepping, lunging/long-stepping, marching. Upper body: BUES, flies (horizontal abduction/adduction), rows, forward press (shoulder flexion/extension), lateral press (shoulder abduction/adduction), biceps/triceps exercises. UBE, forward and backward × 30 s each direction. Note: these were performed with or without resistance paddles, 2 × 10 R each. Lower body: hip movements (flexion, extension, abduction, adduction), heel/toe lifts, squats, step-ups. Note: These were performed with or without ankle weights 2 × 10 R each. Trunk: rotation, side bending, wall pushups. Note: 2 × 10 R each. Cardiovascular activities: jogging in place, jumping jacks, cross-country skiing, hopping (forward/backward, left/right). Note: 30 s for each activity, progressing to 2 sets. Optional activities: stationary swim kicks while prone on kickboard/noodle/pool wall, cycling in noodle.	Endurance ↑ balance ↑ mobility ↑ core body strength ↑ handgrip strength ↔ speed ↔		
Casey A.; Rasmussen R.; Mackenzie S.; Glenn J. (2010) [19]	6 M + 2 F with ID 13.1 ± 3.4 years	Swimming, 60 min, 3 days/week, 16 weeks	Volume per session: 200 m to 500 m to 1000 m, 1 to 2 min rest between sets. Participants focused on performing the front crawl stroke and carrying out exercises with the assistance of a flutter board device. Participants also engaged in 10 min of dry-land training.		Body mass ↔ body fat ↑ BMI ↔	
Pan C.Y. (2010) [24]	8 M with ID 7.3 ± 1.2 years	Adapted swimming, 90 min, 2 days/week, 10 weeks	Social activities and interaction. Warmup (20 min): limbs and truck exercise, splashes water. Main (40 min): water orientation skills, breathing, swimming, floating, and stroke skills. Group activities (20 min): games, social interaction activities (e.g., noodle, kick, jump, float, hula-hoop). Cooldown (10 min).	Aquatics skills ↑ balance ↑		
Fragala-Pinkham, M. A.; Haley, S. M.; O’Neil, M. E. (2011) [23]	7 M + 1 F with DS 9.6 ± 2.6 years	Adapted swimming, 40 min, 2 days/week, 14 weeks	Balls, basketball net, noodles, bar bells, cuff weights. Warmup: running in place, jumping jacks, reciprocal arm, and leg movements, hopping on one foot, karate kicks, jumping in place and jumping forward, backward, and sideways. Swimming laps: front stroke, elementary backstroke, front crawl, back crawl, breaststroke, and kicking with a kickboard. Gamified exercises. Cooldown: marching in place, arm circles, and leg circles. Stretching in the shallow end of the pool.	Mobility skills ↔ cardiorespiratory endurance ↔ muscular endurance ↔	heart rate ↔	
Casey A.; Boyd C.; MacKenzie S.; Rasmussen R. (2012) [22]	6 M + 2 F with ID 43.0 ± 13.9 years	Adapted swimming, 60 min, 3 days/week, 13 weeks	Additional equipment (e.g., aqua jogging belts, flutter boards). Aqua jogging, water polo and lap swimming. The duration of intense exercise increased from 15 to 25 to 35 min at 4-week intervals. Endurance exercise was preceded by a 10 min light aerobic warm-up and ended with a 10 min low-intensity recovery period.		BMI ↔ body fat ↔ heart rate ↔	
Lee I. H.; Seo E. J.; Lim I. S. (2014) [21]	15 M with ID 15.60 ± 2.19 years	Adapted swimming, 50 min, 3 days/week, 12 weeks	Warmup (10 min): stretching in water, front, side, and back walking in water. Main exercise (30 min): kicking, knees to chest, inside and outside movement of feet, running, sidestep, jumping, free swim. Cooldown (10 min): stretching in water, front, side, and back walking in water.			BDNF ↑ IGF-1 ↔ VEGF ↑

↑ increase, ↓ decrease, ↔ unaffected. M—males; F—females; ID—intellectual disabilities; min—minutes; DS—Down syndrome; m—meters; VO2max—maximal oxygen consumption; HRmax—maximum heart rate; s—seconds; BMI—body mass index. R—repetitions; RBR—rest between repetitions; PR—passive recovery; RBS—rest between sets; SS—swimming speed; HRR—maximum heart rate reserve; RPE—rating of perceived exertion; TC—total cholesterol; TG—triglyceride; LDL-C—low-density lipoprotein cholesterol; HDL-C—high-density lipoprotein cholesterol; IgA—immunoglobulin A; IgG—immunoglobulin G; IgM—immunoglobulin M; UES—bilateral upper extremity scaption; UBE—upper body ergometer; R—repetitions; BDNF—brain-derived neuropathic factor; IGF-1—insulin-like growth factor; VEGF—vascular endothelial growth factor.

## 4. Discussion

The aim of this review was to identify research carried out in recent years (2010–2022) on the functional ability and the physiological and biochemical status of people with IDs. One problem that was evident in the literature was the use of interventions by researchers that were complex and not well controlled. Confounding, several reported interventions used exercise protocols that were performed at different temperatures and were similar to those used on dry land. In such cases, participants typically moved along the bottom of the pool. The main aim of this review was to focus on swimming, where exercise is applied when moving across the length of the pool, by means of regular swimming, kicking, or using a pullbuoy to train the arms.

We identified seven studies in which participants swam repeated pool lengths [5,6,12,16,17,18,19]. In five other studies, swimming was part of an aquatic intervention and was only vaguely described [20,21,22,23,24]. Lastly, three studies were found where only aquatic intervention was performed in the form of a dance program [9] or motor activity [10,25].

Monitoring the intensity of the activity is an important element in the evaluation of the intervention [25]. The simplest way of achieving this is to measure the time taken to complete the exercise; this may involve covering a specified distance or performing a set number of repetitions. Another method—made possible by the technology available today—is heart rate monitoring. No study recorded the time taken for swimming repetitions. Some researchers reported that volume of activity (time or distance) increased during the intervention [6,12,19,20,22,23,25]. In the study by Suarez-Villadat et al., heart rate was assessed by placing the index and third fingers on the neck of participants at the side of their trachea [5]. In four studies, heart rate-recording watches or chest straps were used [19,20,22,23].

One issue arising from our review concerns the use of different terminology by researchers for similar parameters such as aerobic capacity, endurance, and cardiorespiratory fitness. Some terms are used generally, such as muscle strength, power, and speed; others are used specifically for changes in parameters such as strength or speed, core body strength, grip strength, and limb speed.

Concerning basic functional abilities such as endurance and strength, researchers [6,12] found that it takes more than 8 weeks of 60 min sessions, performed three times per week, for improvement in these capacities to be observed. Similarly, when considering regular interventions involving conventional swimming, it was observed that more than 8 weeks are required to observe any changes in functional capacity in people with IDs [12]. The authors of [6] found that long-term swimming intervention had a positive effect on endurance by increasing VO_2_ max, while HR max remained unaffected.

In recent years, studies focusing on swimming programs have concentrated only on functional abilities and biological parameters. Improved functional ability may be related to improvements in aerobic capacity and/or muscle strength [12,25]. However, such improvement has also been reported after smaller scale interventions, such as the 10-week water exercise and swimming program, involving 40 min sessions twice a week, reported by the authors of [28]. Contrarily, the 12-week study carried out by Pérez et al. did not find improvements in any functional parameters, probably as a result of the short duration of the swimming repetitions and sets in the intervention [16]. Furthermore, the top study that showed no improvements did not provide detailed information on the swimming intervention involved [18].

In terms of balance, the results were mixed. Terms used in studies included the following: dynamic balance (walking on a balance beam) [12,18,25]; motor balance control skills [6,24]; timed up-and-go [10]; and static balance (standing on one leg) [6,12,25]. Aquatic training and swimming caused different effects. For example, dynamic balance and motor balance control skills in water either improved [6,10,12,24] or remained unchanged [18,25]. However, no improvements were reported with respect to static balance [6,12,25].

Differing effects on body mass were reported in the studies, regardless of the duration of the intervention. However, there was no recording of the participants’ diets to support these results. One study proposed the hypothesis that lower water temperature increases energy expenditure, leading to improvements in body mass [12]; another study highlighted the extreme difficulty of monitoring and controlling dietary intake [19]. Researchers also found that it took a high volume of intervention to reduce BMI [5,6,12]; this was achieved, of course, by reducing body mass. Regarding body fat, some studies found that a high volume of intervention had a positive effect, in terms of decreased body fat [5,6]; another study found that a lower volume of intervention combined with difficulties in controlling energy intake led to an increase in body fat [19].

Only three studies considered biochemical markers. Swimming for 60 min, 2 days per week, for 16 weeks, did not appear to significantly alter biochemical parameters such as TC, LDL-C, HDL-C, IgA, or IgM; however, such an intervention did lead to increased levels of TG and IgG [17]. Any changes in these parameters suggest an important role for seasonal factors such as the effect of cold weather rather than athletic activity [29]. Reductions in IgA and IgG require exercise of high volume and high intensity, similar to that of a triathlon event [30]. Salivary cortisol decreased after 12 weeks of aquatic training involving 40 min of SWAN, once per week [9]. Diurnal cortisol variation in subjects with ID followed the pattern of subjects in the general population [9]. Aquatic intervention resulted in reduced cortisol levels over time, as well as lower recorded levels after some individual sessions. This suggests that the intervention contributed to stress reduction. On the other hand, it should be noted that SWAN performed in a warm pool (33–35 °C) was not evaluated for intensity [9]. A contributing parameter to stress reduction could be water temperature, as it has been shown that, after cycling in a 30–35 °C water environment, cortisol levels decreased [31].

BDNF and VEGF increased but IGF-1 was unaffected by a 12-week intervention of adapted swimming for 50 min, 3 days per week, in which the main exercises were 30 min of kicking, knees to chest, inside and outside movement of feet, running, sidestep, jumping, and free swimming [21]. The findings of the studies reviewed here suggest that exercise boosts these growth factors, leading to improved overall brain health [32,33]; however, in one recent study similar to that of Lee et al. [21], involving loading characteristics that were nearly 75% of maximal HR, with 25–40 min of exercise for 12 weeks, there appeared to be no significant changes in circulating plasma levels of BDNF, VEGF165, and IGF-1 in adults with Alzheimer’s disease [34]. This suggests the need for further research on these critical biochemical factors, which considers the intensity of exercise expressed mainly in terms of heart rate.

## 5. Conclusions

The most troublesome aspect of our review concerns the incomplete recording of data on exercise intensity and the precise timing of physical activity, as well as the wide variation in terminology used to describe physical abilities. However, the study results indicated that aquatic interventions led to improvement in endurance and strength, although the evidence was conflicting for balance and body mass index, and there was only limited evidence for biomarkers related to stress and brain health. Further research involving conventional swimming is needed, in which data such as heart rate measurements, the time taken to cover specific distances, and water temperature are recorded. Overall, the framework of physical activity for people with IDs needs to be made clearer so that physical fitness, biological parameters, and biochemical parameters can be better assessed and enriched with more data. This will contribute to the effort to improve the physical fitness and mental health of these vulnerable people.

## Figures and Tables

**Figure 1 healthcare-11-01990-f001:**
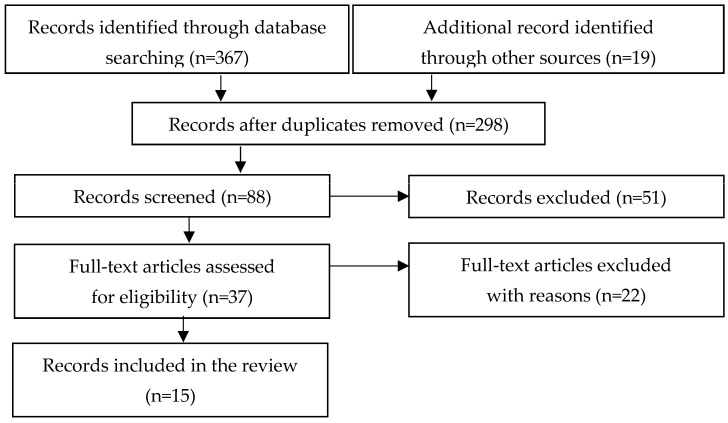
PRISMA, diagram of preferred reports for reviews.

**Table 1 healthcare-11-01990-t001:** The six keyword combinations in this review.

Keywords
“intellectual disability” OR “mental retardation” AND “physical activity” OR “swimming”
“intellectual disability” OR “mental retardation” AND “aquatic exercise” OR “swimming”
“intellectual disability” OR “mental retardation” AND “aquatic exercise” OR “swimming” AND “strength” OR “power” AND “balance”
“intellectual disability” OR “mental retardation” AND “aquatic exercise” OR “swimming” AND “body composition” AND “fat mass”
“intellectual disability” OR “mental retardation” AND “aquatic exercise” OR “swimming” AND “hematological parameters” OR “biochemical parameters”
“intellectual disability” OR “mental retardation” AND “aquatic exercise” OR “swimming” AND “stress” OR “benefits”

OR—the disjunction between the terms; AND—the conjunction between the terms.

## Data Availability

Not applicable.

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
