# Peer review of "Effects of Aquatic Interventions on Physical Health Indicators in People with Intellectual Disabilities: A Review"

_healthcare, 2023, doi:10.3390/healthcare11141990_

Round 1
Reviewer 1 Report
The selected topic, aquatic interventions, is a potentially interesting one and the results could inform future research and clinical practice. There are, however, weaknesses in the paper, some of which are due to the writing style, others to the content.
Line 17. to be clearer, "accurate load data" should include what type of "load" is addressed
Line 23. a little more detail on why those with IDs have special needs/concerns about aquatic and other forms of physical activity is needed
Line 27 the word "care" is missing
Line 34. reword so that this does not sound as if people with ID are not human beings
Line 65. it does not appear that all of the studies selected for review are actually longitudinal. Further, no reason is given for combining studies on children and adults in one review. This is unusual and needs to be justified.
Line 72. do you mean for the process or for the outcome?
Line 93. Please be more specific than saying "almost" all of the studies
Line 136. the table is difficult to read quickly with one portion vertical and the other horizontal. Please consult with editor re appropriate formatting.
Lines 97, 103. Please explain the distinction you have made between aerobic capacity and cardiorespiratory fitness here. Note that later you criticize the way these terms are used by others.
Line 171. here is where you (correctly) critique use of different terminology for similar phenomena
Line 183. How are the findings from these studies different from those in other studies of children and adults without ID who are involved in aquatic activities?
Line 212. Please explain "seasonal" factors
Line 236. These are good points regarding incomplete data in the studies reviewed. In the Discussion and Conclusion sections overall, however, a thorough discussion of the implications of the findings for practice and future research are missing.
The paper is readable but there are instances where it is not clear if the authors have chosen the correct term, ie, if that is what was meant.
Author Response
Dear reviewer,
thank you for taking the time to review our work, it is much appreciated. We have attempted to address your questions in more detail, as below. Unfortunately, the line numbers on the pages mentioned in your comments do not correspond to ours in the downloaded manuscript.
Comments and Suggestions for Authors
The selected topic, aquatic interventions, is a potentially interesting one and the results could inform future research and clinical practice. There are, however, weaknesses in the paper, some of which are due to the writing style, others to the content.
Line 17. to be clearer, "accurate load data" should include what type of "load" is addressed
Corrected to "training load"
Line 23. a little more detail on why those with IDs have special needs/concerns about aquatic and other forms of physical activity is needed.
We added two sentences in the paragraph:
People with IDs experience physical problems exacerbated by low fitness levels, including functional parameters such as strength, endurance, coordination, and balance. The lack of physical abilities makes it difficult for these people in their daily lives, such as moving around and looking after themselves. Also, many health problems occur at high rates in people with intellectual disability, such as specific syndromes, for example, congenital heart defects, poor mental health, and psychosocial disturbances. For these reasons, in addition to physical condition, other healthcare indicators, such as biological and biochemical parameters, should be considered when monitoring the vital status of people with IDs. Data on body composition, heart rate, blood lipids, and hormones are all necessary to fully evaluate biological function in such individuals [4,5].
Line 27 the word "care" is missing
Line 29. “health” changed to healthcare (Line 34). I hope I put the word correctly if that's what you meant. I don't understand why it's missing from line 27
Line 34. reword so that this does not sound as if people with ID are not human beings
Dear reviewer, we rephrase the quoted sentence to “It is well known that regular physical activity contributes to maintaining and improving the functional abilities in people with or without IDs [6,7].”
Line 65. it does not appear that all of the studies selected for review are actually longitudinal. Further, no reason is given for combining studies on children and adults in one review. This is unusual and needs to be justified.
For the first observation: The shortest study was six weeks, and the longest was six months. For this reason, we preferred the term longitudinal. We also provide a web reference.
How long is a longitudinal study?
No set amount of time is required for a longitudinal study, so long as the participants are repeatedly observed. They can range from as short as a few weeks to as long as several decades. However, they usually last at least a year, oftentimes several.
https://www.scribbr.com/methodology/longitudinal-study/
For the second observation: Because the aquatic interventions for people with ID have been limited over the last decade, we wanted to highlight all of them. We wondered why there is a large and varied reporting on physical abilities, further to a lack of results on biochemical health parameters. Exercise also affects in the same direction children and adults, although to different degrees of adaptation.
We cite here the review of "Salse-Batán et al. (2023) Aquatic exercise for people with intellectual disabilities: findings from a systematic review, International Journal of Developmental Disabilities, 69:2, 134-146, DOI: 10. 1080/20473869.2021.1924033" in which they state that "We should also add that the included articles differed in terms of the age of the participants and the severity of the intellectual disability (with the latter not always explicitly specified)." Unfortunately, this systematic review did not include in our references.
Line 72. do you mean for the process or for the outcome?
Line 76. We meant “process”
Line 93. Please be more specific than saying "almost" all of the studies
Line 97. Changed to “Most of the studies in our review”
Line 136. the table is difficult to read quickly with one portion vertical and the other horizontal. Please consult with editor re appropriate formatting.
We will do it
Dear reviewer, we inform you that we have created an alternative form in the results tables at the end of the manuscript.
Lines 97, 103. Please explain the distinction you have made between aerobic capacity and cardiorespiratory fitness here. Note that later you criticize the way these terms are used by others.
These terms were given as reported by the researchers in their studies, so we critiqued them. We grouped the results and reported them as endurance in the discussion (line 251).
Line 171. here is where you (correctly) critique use of different terminology for similar phenomena
Thank you very much for your observation. Thank you for confirming our idea of grouping basic physical abilities.
Line 183. How are the findings from these studies different from those in other studies of children and adults without ID who are involved in aquatic activities?
The literature suggests that the results are in the same direction, provided that the aquatic interventions are implemented over a long period.
- Mitic, M.; Aleksandrovic, M. Effects of physical exercise on motor skills and body composition of adults with intellectual disabilities: A systematic detailed review. J. Anthropol. Sport Phys. Educ. 2021, 5, 27–34. doi.org/10.26773/jaspe.211005
- Cugusi, L.; Carta, M. G. Conventional exercise interventions for adults with intellectual disabilities: A systematic review and meta‐analysis. Transl. Sports Med. 2020, 4, 6–20. doi.org/10.1002/tsm2.195
- Bouzas, S.; Martínez-Lemos, R. I.; Ayán, C. Effects of exercise on the physical fitness level of adults with intellectual disability: A systematic review. Disabil. Rehabil. 2018, 41, 3118–3140. doi.org/10.1080/09638288.2018.1491646
Line 212. Please explain "seasonal" factors
The phrase "seasonal factors such as the effect of cold weather" has been added
Line 236. These are good points regarding incomplete data in the studies reviewed. In the Discussion and Conclusion sections overall, however, a thorough discussion of the implications of the findings for practice and future research are missing.
Dear reviewer, in the conclusion paragraph, the implications of the present review reported "The most troublesome aspect of our literature review concerns the incomplete recording of data on exercise intensity and the precise timing of physical activity, as well as the wide variation in terminology used to describe physical abilities.” Also, it is recommended: “Further research involving conventional swimming is needed, in which data such as heart rate measurements, the time taken to cover specific distances, and water temperature are recorded.”
.Comments on the Quality of English Language
The paper is readable but there are instances where it is not clear if the authors have chosen the correct term, ie, if that is what was meant.
Dear reviewer, the paper has undergone English language editing by MDPI (English-Editing-Certificate-66155).

Reviewer 2 Report
Hello Dears;
Thanks for the useful and practical article.
Comments:
1-In the studies conducted, the age range of patients with I.D has not been mentioned.
2- The diagnostic criteria of M.R and I.D are not exactly the same.
Mental Retardation(M.R) in DSM-IV mentions the reduction of I.Q but Intellectual Disability (I.D) means reduction of I.Q and adjustment disorder(both of them).
Author Response
Dear reviewer,
thank you for taking the time to review our work, it is much appreciated. We have attempted to address your questions in more detail, as below.
Comments and Suggestions for Authors
Thanks for the useful and practical article.
Comments:
1-In the studies conducted, the age range of patients with I.D has not been mentioned.
The age range of patients with ID is reported in Tables 2-5.
2- The diagnostic criteria of M.R and I.D are not exactly the same.
Mental Retardation (M.R) in DSM-IV mentions the reduction of I.Q but Intellectual Disability (I.D) means reduction of I.Q and adjustment disorder (both of them).
Dear reviewer, we agree with your observation. For the convenience of readers, we decided to use the term "Intellectual Disability (ID)" based on the analysis of Carulla, et al. [3], who replaced the terminology "Intellectual Disability" with "Intellectual Developmental Disorders (IDD)" (p. 175) and conclude that “The name and definition of IDD proposed by the Working Group does not conflict with the use of ID terminology” (p. 178).

Reviewer 3 Report
Thank you for the opportunity to review this manuscript called: Effect of Aquatic Intervention on Physical Health Indicators in People with Intellectual Disability: A Literature Review
The authors have put a good amount of work into this paper, for which they are thanked, however, there are still several areas that require critical revision before the paper will likely be strengthened and of interest to an international audience.
The paper would benefit from grammatical revision. I will not list many suggestions in relation to this but highly recommend that the paper undergo further revision prior to sending it out for further review. I will concentrate on method-related issues and areas that need development in the analyses and how they are linked to the conclusion.
ABSTRACT:
The INTRODUCTION is fairly vague.
AIMS: The aims of the paper were to conduct a systematic review that investigates the effectiveness of Aquatic Intervention on Physical Health Indicators in People with Intellectual Disability. There is no mention in the aims of whether this study relates to nursing generally, so this needs further refinement.
METHODS: There is no research question developed with a PICO or other common methods for lit. reviews and no clear search strategy provided.
I note that three databases are used. Why you selected only these three database? Also, Researchgate and goggle scholar are not formal research databases
METHODS: There is no discussion about what type of framework the authors used to conduct the systematic review – e.g. PRISMA, JBI…?
There is no research question fiting within an overarching research question, such as might be developed using PICO or similar frameworks. Suggest a review of this and an addition of a well-developed research question to help guide concepts that the keywords will fit into for the search.
ELIGIBILITY CRITERIA: Which articles were included – was only original research, or did the authors include secondary research? Were these developed a priori?
Why focusing on the last 13 years? It seems like a large body of literature may be lost. This seems to be done purely for convenience and I doubt it will result in a better or more convincing piece of work.
The search strategy is not formal and is thus not replicable. I would doubt the authors could get to the same papers if they followed the description on their paper. This is an example of an appropriate search strategy: https://guides.lib.umich.edu/c.php?g=283340&p=2126706
EXCLUSION CRITERIA: were conference abstracts excluded?
SEARCH STRATEGY: Were tools such as Covidence and EndNote (or another referencing tool) used? It is stated that the initial search used all keywords independently and that only resulted in very few papers – this seems very unlikely and more information is needed to demonstrate the authors’ research strategy (e.g. included in a table, with keywords grouped by concepts and linked by Boolean operators and wildcards, etc). How was deduplication performed? How many researchers/authors were there and who sorted out discrepancies amongst them?
Where is THE PRISMA FLOW CHART?
DESCRIPTION OF THE SELECTED STUDIES: More information is needed about the studies and their findings.
DISCUSSION. Some references are needed to support the discussion or even a reference to the summary table.
What are the study implications and recommendations for future research?
TABLE 1 . All abbreviations that appear in the table need to be in the notes under the table.
Thank you and good luck with your work. Please understand that my multiple suggestions are only provided to help to improve the study and are in no way meant to disparage
Moderate editing of English language required
Round 2
Reviewer 1 Report
Many of the major concerns have been addressed. A few more remain:
Use of the term "mini" review downgrades the importance of what is reported and conflicts with description as a systematic review in line 255. The term is inappropriate and should be discarded. Who would want to read a "mini" review?
Although you have found a definition of "longitudinal" that fits your use, you are diminishing the term with this use and it is unnecessary. Once a study is experimental, by definition it includes at least two time points. Suggest deleting the term.
Line 171 Please spell out PICO the first time it is used.
Just a few uses that are arguable, may be more a reflection of their thinking than of English usage.
Author Response
Dear reviewer,
thank you for taking the time to review our work, it is much appreciated. We have attempted to address your questions in more detail, as below.
- Use of the term "mini" review downgrades the importance of what is reported and conflicts with description as a systematic review in line 255. The term is inappropriate and should be discarded. Who would want to read a "mini" review?
We will keep the phrase “a review” throughout the manuscript.
- Although you have found a definition of "longitudinal" that fits your use, you are diminishing the term with this use and it is unnecessary. Once a study is experimental, by definition it includes at least two time points. Suggest deleting the term.
DONE
- Line 171 Please spell out PICO the first time it is used.
DONE
- Comments on the Quality of English Language
Just a few uses that are arguable, may be more a reflection of their thinking than of English usage.
Sorry for this misquote of our thoughts in English.
Once again thank you for the important help

Reviewer 3 Report
Thank you for your effort on the revised manuscript. However, I still have comments:
The format Prisma flow chart is not the most updated.
If you only focus on swimming in this review, why is this not reflected in the title?
How is your review different from the review of Cugusi [8], Grosse [11], and Bouzas [13]?
Minor editing of English language required
Author Response
Dear reviewer,
thank you for taking the time to review our work, it is much appreciated. We have attempted to address your questions in more detail, as below.
Thank you for your effort on the revised manuscript. However, I still have comments:
- The format Prisma flow chart is not the most updated.
Dear reviewer, we changed the flow chart according to Mitic et al. (2021) [7].
- If you only focus on swimming in this review, why is this not reflected in the title?
Dear reviewer, this was the first idea we had. But when fewer studies included swimming, we used "aquatic" in the title. If you insist, we will change it to "swimming interventions."
- How is your review different from the review of Cugusi [8], Grosse [11], and Bouzas [13]?
Dear reviewer, first, it must be mentioned that this review started a long time ago, as our interest was in swimming (first author) and swimming training of people with intellectual disabilities (second author). Subsequently, our review focused on swimming specifically. The difference from the Cugusi and Bouzas reviews was their findings that the physical fitness of people with intellectual disabilities might have been affected by land-based exercises. They also did not include papers where biological and biochemical parameters related to physical and mental health were examined. Finally, Grosse's review focuses on recording the studies and their purposes without summarizing their results.
- Comments on the Quality of English Language
Minor editing of English language required
We will do it.
Once again thank you for the important help
